

# Mitochondrial diversity in *Gonionemus* (Trachylina:Hydrozoa) and its implications for understanding the origins of clinging jellyfish in the Northwest Atlantic Ocean

Annette F. Govindarajan[1], Mary R. Carman[1], Marat R. Khaidarov[2,3], Alexander Semenchenko[3] and John P. Wares[4,5]

[1] Biology Department, Woods Hole Oceanographic Institution, Woods Hole, United States
[2] A.V. Zhirmunsky Institute of Marine Biology, National Scientific Center of Marine Biology Far Eastern Branch, Russian Academy of Sciences, Vladivostok, Russia
[3] Far Eastern Federal University, Vladivostok, Russia
[4] Department of Genetics, University of Georgia, Athens, United States
[5] Odum School of Ecology, University of Georgia, Athens, United States

Corresponding author
Annette F. Govindarajan,
afrese@whoi.edu

## ABSTRACT

Determining whether a population is introduced or native to a region can be challenging due to inadequate taxonomy, the presence of cryptic lineages, and poor historical documentation. For taxa with resting stages that bloom episodically, determining origin can be especially challenging as an environmentally-triggered abrupt appearance of the taxa may be confused with an anthropogenic introduction. Here, we assess diversity in mitochondrial cytochrome oxidase I sequences obtained from multiple Atlantic and Pacific locations, and discuss the implications of our findings for understanding the origin of clinging jellyfish *Gonionemus* in the Northwest Atlantic. Clinging jellyfish are known for clinging to seagrasses and seaweeds, and have complex life cycles that include resting stages. They are especially notorious as some, although not all, populations are associated with severe sting reactions. The worldwide distribution of *Gonionemus* has been aptly called a "zoogeographic puzzle" and our results refine rather than resolve the puzzle. We find a relatively deep divergence that may indicate cryptic speciation between *Gonionemus* from the Northeast Pacific and Northwest Pacific/Northwest Atlantic. Within the Northwest Pacific/Northwest Atlantic clade, we find haplotypes unique to each region. We also find one haplotype that is shared between highly toxic Vladivostok-area populations and some Northwest Atlantic populations. Our results are consistent with multiple scenarios that involve both native and anthropogenic processes. We evaluate each scenario and discuss critical directions for future research, including improving the resolution of population genetic structure, identifying possible lineage admixture, and better characterizing and quantifying the toxicity phenotype.

## INTRODUCTION

Invasive species can have harmful impacts on native taxa and disrupt ecosystem functioning (*Molnar et al., 2008*; *Gallardo et al., 2016*) and in some cases have direct negative impacts

on human health (*Ruiz et al., 2000*; *Pyšek & Richardson, 2010*). Reconstructing invasion histories is important for illuminating anthropogenic dispersal pathways and management (*Molnar et al., 2008*). However, determining if and when species invasions have occurred is challenging for several reasons: the invasion may have occurred decades or more in the past, local taxonomic baseline data may be lacking, and the invaders may be morphologically similar to local organisms (*Carlton, 2009*). Compounding these difficulties, there may be multiple inputs of invaders, sometimes originating from different source regions (*Geller et al., 1997*; *Simon-Bouhet, Garcia-Meunier & Viard, 2006*; *Roman, 2006*). Accordingly, many species are considered cryptogenic and cannot be classified as introduced or native with confidence (*Carlton, 1996*). In some instances, organisms may exhibit different phenotypes in their introduced range than in their native range (*Miglietta & Lessios, 2009*; *Krueger-Hadfield et al., 2016*), making morphology-based identifications difficult. Molecular approaches can be very useful in identifying non-native taxa (*Geller, Darling & Carlton, 2010*), but results are also not always clear-cut. Sufficient sampling in both the source and invasive ranges is crucial to accurately resolve invasion histories and dispersal pathways (e.g., *Darling et al., 2008*; *Yund, Collins & Johnson, 2015*).

The hydrozoan *Gonionemus vertens* (A. Agassiz in *Agassiz, 1862*) is native to the Pacific and thought to be introduced in Europe (*Edwards, 1976*; *Bakker, 1980*), eastern North America (reviewed in *Govindarajan & Carman, 2016*), and South America (*Rodriguez et al., 2014*). Its worldwide distribution has been called a "zoogeographic puzzle" due to its episodic and disjunct populations (*Tambs-Lyche, 1964*). In the Northwest Atlantic, *G. vertens* was first recorded in 1894 in Woods Hole, MA and nearby locations. Medusae typically cling to eelgrass and macroalgae in sheltered shallow water bays and coves. *G. vertens* life cycle includes cryptic polyp and cyst stages (*Perkins, 1902*; *Uchida, 1976*) which could facilitate its unintentional dispersal to new areas. In Woods Hole, medusae abruptly disappeared circa 1930 when the eelgrass, its preferred substrate, was decimated due to a wasting disease (*Bakker, 1980*). However at least one small population persisted on the nearby island of Martha's Vineyard (approximately 8 km south of Woods Hole) (*Govindarajan & Carman, 2016*). As well, there were sporadic medusae sightings over several decades in the Gulf of Maine (*Govindarajan & Carman, 2016*). Beginning in 1990, *G. vertens* appeared to experience a regional resurgence, with regular new observations near and beyond the Woods Hole region in the Northwest Atlantic (*Govindarajan & Carman, 2016*).

*G. vertens* is notorious for causing severe stings in the Northwest Pacific in the Sea of Japan (*Pigulevsky & Michaleff, 1969*; *Yakovlev & Vaskovsky, 1993*). Symptoms vary between victims, but may include extreme pain, respiratory distress, paralysis, hallucinations, and blindness, which can last 3–5 days. *G. vertens* in the eastern Pacific and until recently, in its invasive ranges, has not been reported to cause sting reactions in humans (*Naumov, 1960*). However, since 1990, some Northwest Atlantic populations appear to cause stings similar to those reported from the western Pacific (*Govindarajan & Carman, 2016*). These observations, coupled with regular new sightings, suggest that a second wave of *G. vertens* invaders, originating from the western Pacific, could be present in the Northwest Atlantic (*Govindarajan & Carman, 2016*).

Recognition of introduced species can be hampered by inadequate taxonomy (*Carlton, 2009*). *G. vertens* was originally thought to comprise two varieties, *G. vertens vertens* and *G. vertens murbachii* Mayer 1901 (*Naumov, 1960*). *G. vertens vertens*, with a "hemisphaerical or somewhat flattened" umbrella and faint yellow-green coloration, was described from the eastern and western North Pacific, including in the Sea of Japan, the Aleutians Islands, and Puget Sound. *G. vertens murbachii*, with a completely transparent umbrella and numerous tentacles, was thought to occur in the Atlantic coasts of Europe and North America and the Mediterranean (*Naumov, 1960*). *Russell (1953)* suggested that *G. murbachii* may be characterized by a flatter bell than *G. vertens*. Dangerous stings were recorded from a subset of western Pacific *G. vertens vertens* only (*Naumov, 1960*). Later thinking viewed the Atlantic forms as stemming from the Pacific via human-mediated transport (*Tambs-Lyche, 1964*; *Edwards, 1976*; *Bakker, 1980*; *Govindarajan & Carman, 2016*). Because the initial Atlantic forms did not cause painful stings, it was thought that they originated from the less toxic northeastern Pacific populations. Historical northwest Atlantic *G. vertens* were believed to have arrived either directly from the northeastern Pacific, or indirectly by way of Europe (*Edwards, 1976*), while contemporary *G. vertens* is hypothesized to have arrived from a northwestern Pacific source (*Govindarajan & Carman, 2016*).

Our goal is to determine the origin of contemporary Northwest Atlantic (NWA) *Gonionemus* populations. We will test the hypothesis that contemporary NWA populations derive some diversity from Northwest Pacific (NWP) populations. To better constrain possible sources, we will also assess population divergence between Northwest Pacific (NWP) and Northeast Pacific (NEP) populations. We evaluate 182 mitochondrial cytochrome oxidase I (COI) sequences from *Gonionemus* obtained from 12 North Atlantic and potential source populations in the North Pacific. Our sampling includes NWA sites where *Gonionemus* has not previously been recorded. It also includes NWP sites that harbor highly toxic populations in the Peter the Great Gulf near Vladivostok, Russia.

## METHODS

### Sample collection

Atlantic and Pacific medusae were collected by snorkeling or by net from boats or floating docks, or from colleagues (Table 1). We refer to locations in the Northwest Atlantic as NWA, Northeast Atlantic as NEA, Northwest Pacific as NWP, and Northeast Pacific as NEP. Medusae were either used live or preserved in ethanol for later DNA processing.

### Molecular methods

For the mitochondrial DNA assessment, genomic DNA was extracted using DNEasy Blood & Tissue kits (Qiagen) following the manufacturer's instructions. An approximately 600 base pair part of the mitochondrial COI gene was amplified using universal primers (*Folmer et al., 1994*) under the following conditions: 95° 3 min, 35 cycles of 95° for 30 s, 48° for 30 s, 72° for 1 min, and a final cycle of 72° for 5 min. PCR products were visualized on a 1.2% agarose gel stained with GelRed (Biotium) and purified using Qiaquick PCR purification kits (Qiagen). Products were quantified with a Nanodrop 2000 spectrophotometer (Nanodrop Technologies, Wilmington, Delaware, USA) and sequenced

**Table 1  Sampling locations and location abbreviations.**

| Region | Site | Abbreviation | Latitude | Longitude | Reference |
|--------|------|--------------|----------|-----------|-----------|
| NWA | Great Bay, NH | GB | 43.070 | −70.900 | This study |
| | Bass River, Yarmouth, MA | BR | 41.707 | −70.186 | This study |
| | Hamblin Pond, Mashpee, MA | HP | 41.573 | −70.508 | This study |
| | Farm Pond, Oak Bluffs, MA | FP | 41.447 | −70.557 | This study |
| | Sengekontacket Pond, Edgartown, MA | SG | 41.416 | −70.569 | This study |
| | Potter Pond, North Kingston, RI | PP | 41.382 | −71.531 | This study |
| | Mumford Cove, Groton, CT | MC | 41.324 | −72.019 | This study |
| | Pine Island, Groton, CT | PI | 41.314 | −72.058 | This study |
| NWP | Amur Bay (Peter the Great Gulf) | AB | 43.199 | 131.919 | This study |
| | Vostok Bay (Peter the Great Gulf) | VB | 42.892 | 132.729 | This study |
| | Okirai Bay, Japan (Pacific) | JP | 39.092 | 141.833 | This study |
| | China | CH | N/A | N/A | J He et al., 2013, unpublished data; KF926130–KF926139 |
| NEP | San Juan Island, WA | FH | 48.535 | −123.008 | This study |
| NEA | Álftanes, Iceland | IC | 64.098 | −22.033 | This study |

in both directions by MWG Eurofins Operon. Chromatograms were assembled and DNA sequences were analyzed using the Geneious R9 software platform (*Kearse et al., 2012*). Sequences were aligned using ClustalW (*Larkin et al., 2007*). The alignment was verified by eye, and the ends were trimmed so that all sequences were the same length. In order to facilitate direct comparisons with other published studies (e.g., *Ortman et al., 2010*; *Zheng et al., 2014*), mean Kimura 2-parameter (K2P) distances between locations were calculated with MEGA 7 (*Kumar, Stecher & Tamura, 2016*) and a neighbor-joining haplotype tree based on K2P distances was constructed using the PAUP* 4 (*Swofford, 2003*) plugin in Geneious. We confirmed that the K2P haplotype tree was consistent with a haplotype tree based on the best-fit model (GTR+I, as determined in ModelTest using the Akaike Information Criterion in Geneious; *Posada & Crandall, 1998*). DNA summary statistics were calculated with DnaSP 5.10.01 (*Librado & Rozas, 2009*).

## RESULTS

Samples were obtained from 8 NWA locations ranging from eastern Long Island Sound (near Groton, Connecticut) in the south to Great Bay, New Hampshire in the north; 3 NWP locations (Amur Bay and Vostok Bay in Peter the Great Gulf, Russia and Okirai Bay, Japan); one NEP location (San Juan Island, Washington); and one NEA location (Iceland) (Table 1). Notably, the NWP Amur Bay and Vostok Bay sites are notorious for harboring *Gonionemus* that cause severe stings. While we did not quantitatively assess morphological differences of medusae between regions, we report on some incidental observations. In general, the NWP and NWA samples appeared relatively flat, ranged from dull orange to brown in color, and had relatively diminutive gonads, although there was some variation in all of these traits (Fig. 1). An image of a NWA *Gonionemus* medusa obtained from the Yale Peabody Museum originating from 1965 (pre-sting) appeared similar (although has few

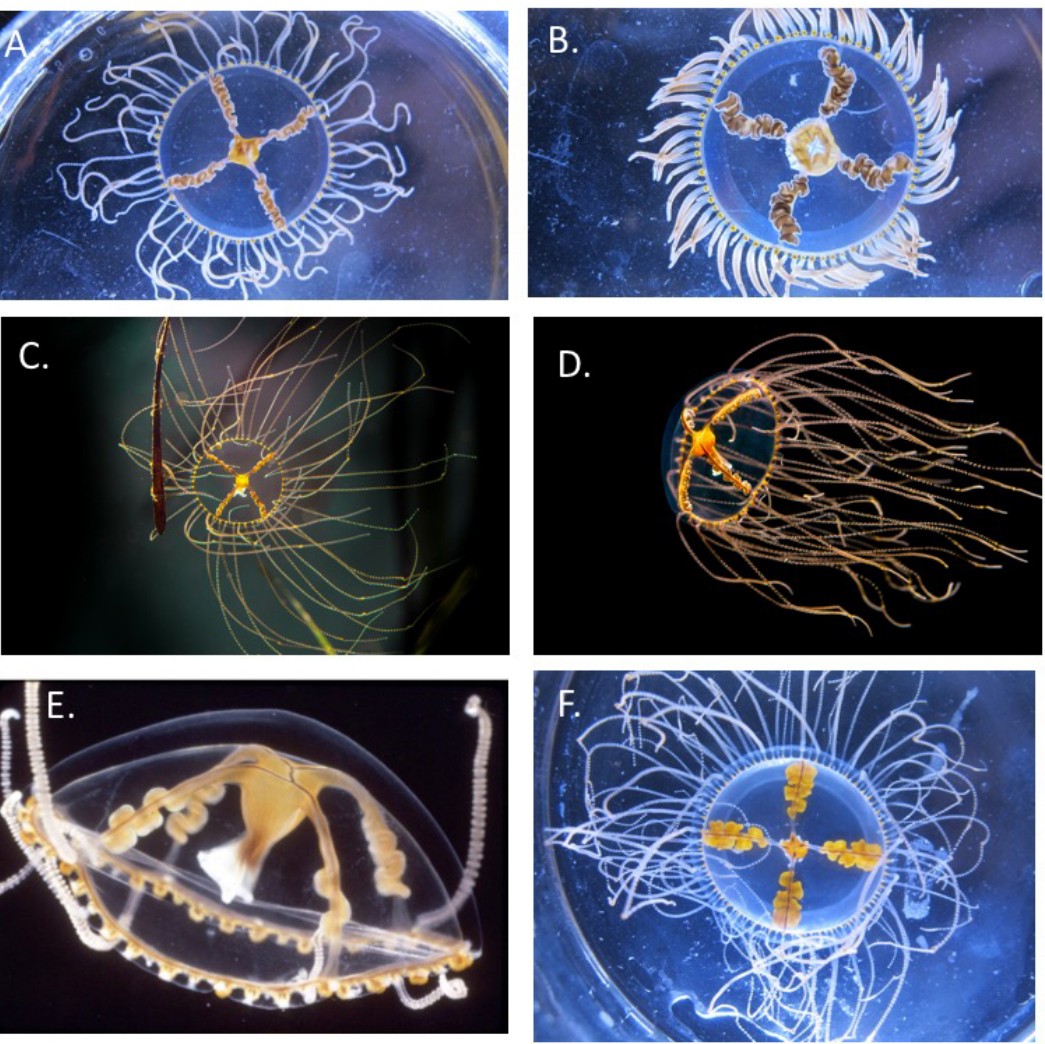

**Figure 1 Comparison of *Gonionemus* medusae.** In general, most of the NWA and NWP medusae observed in this study were relatively flat and have thin, dull brown and orange gonads, and the 4 NEP medusae used in this study were relatively hemispherical and had bright orange fleshy gonads. Gonads are found on the radial canals. Note, photos were taken under different lighting conditions so the colors are not directly comparable. While scale bars are not shown, the maximum size of mature medusae that we recorded was 2.5 cm in the NWA and 3.0 cm in the NWP. (A) Typical contemporary NWA *Gonionemus*. Photo credit A. Govindarajan. (B) Less commonly observed contemporary NWA *Gonionemus* with fleshier gonads. Note the tentacles are contracted. Photo credit A. Govindarajan. (C) NWP (Vladivostok-area) *Gonionemus* with an eelgrass blade. Photo credit L. Petrova. (D) NWP (Vladivostok-area) *Gonionemus* in flow conditions. Photo credit L. Petrova. (E) 1960s NWA *Gonionemus* from the Woods Hole region. Some tentacles are missing but note the relatively thin gonads. Photo credit William Amos, Marine Biological Laboratory, courtesy of the Peabody Museum of Natural History, Yale University. (F) Contemporary *Gonionemus* from the NEP (San Juan Island). Photo credit A. Govindarajan.

**Table 2  Summary statistics for NWA and NWP samples.** Site abbreviations are provided in Table 1.

| Region | Site | N | H | Haplotype counts | Hd | Pi |
|---|---|---|---|---|---|---|
| NWA | GB | 7 | 2 | Hap IV—3<br>Hap V—4 | 0.571 | 0.00912 |
| | BR | 17 | 3 | Hap IV—2<br>Hap V—3<br>Hap VI—12 | 0.485 | 0.01004 |
| | HP | 18 | 1 | Hap VI—18 | 0 | 0 |
| | SG | 3 | 2 | Hap IV—2<br>Hap VI—1 | 0.667 | 0.00931 |
| | FP | 17 | 2 | Hap IV—1<br>Hap VI—16 | 0.0118 | 0.00164 |
| | PP | 22 | 2 | Hap IV—2<br>Hap VI—20 | 0.173 | 0.00242 |
| | MC | 14 | 2 | Hap IV—13<br>Hap VI—1 | 0.143 | 0.00200 |
| | PI | 24 | 2 | Hap IV—23<br>Hap VI—1 | 0.083 | 0.00116 |
| NWP | AB | 3 | 1 | Hap IV—3 | 0 | 0 |
| | VB | 30 | 2 | Hap III—4<br>Hap IV—26 | 0.239 | 0.00048 |
| | JP | 12 | 2 | Hap II—11<br>Hap IV—1 | 0.167 | 0.001 |
| | CH | 10 | 1 | Hap I—10 | 0 | 0 |
| NEP | FH | 4 | 1 | Hap VII—4 | 0 | 0 |
| NEA | IC | 1 | 1 | Hap VII—1 | 0 | 0 |

Notes.

$N$, number of samples; $H$, number of haplotypes; Hd, haplotype diversity; Pi, nucleotide diversity.

tentacles) to contemporary NWA and NWP medusae (Fig. 1). In contrast, NEP medusae were distinctly more hemispherical, with bright orange fleshy gonads (Fig. 1).

We obtained 172 new COI sequences, which were deposited in Genbank (Accession numbers KY437814–KY437985; Table 1). An additional 10 sequences already on Genbank from the western Pacific were also analyzed (Table 1) so our final alignment contained 182 sequences. Our trimmed alignment consisted of 501 base pairs. Overall, we found 7 haplotypes, with the number of haplotypes in a given population ranging from 1 to 3 (Table 2). The mean between-location Kimura 2-parameter (K2P) distances for all locations ranged from 0 to 0.076 (Table 3). Pairwise distances between NWP and NWA locations ranged from 0.001–0.020, while pairwise distances between NWP/NWA and NEP/NEA locations ranged from 0.071–0.076. Haplotype frequencies varied with geographic location (Table 2 and Figs. 2 and 3). Haplotype I was present only in the Chinese specimens, and Haplotype II was present only in Okirai Bay, Japan, and Haplotype III was present only in Vostok Bay. Haplotype IV was found in Vostok Bay and Amur Bay, rare in Okirai Bay, and in varying frequencies in many NWA locations. In the NWA, Haplotype IV was most abundant haplotype in the Pine Island, Mumford Cove, and Sengekontacket Pond locations. It was present, but less abundant, in the Great Bay, Bass River, Farm Pond and

**Table 3   Mean between-location Kimura 2-parameter distances.** Site abbreviations are listed in Table 1.

|    | GB | BR | HP | SG | FP | PP | MC | PI | AB | VB | JP | CH | FH |
|----|----|----|----|----|----|----|----|----|----|----|----|----|----|
| BR | .017 |  |  |  |  |  |  |  |  |  |  |  |  |
| HP | .021 | .006 |  |  |  |  |  |  |  |  |  |  |  |
| SG | .013 | .011 | .009 |  |  |  |  |  |  |  |  |  |  |
| FP | .021 | .007 | .001 | .009 |  |  |  |  |  |  |  |  |  |
| PP | .020 | .007 | .001 | .009 | .002 |  |  |  |  |  |  |  |  |
| MC | .010 | .012 | .013 | .005 | .012 | .012 |  |  |  |  |  |  |  |
| PI | .010 | .013 | .014 | .005 | .013 | .012 | .002 |  |  |  |  |  |  |
| AB | .009 | .013 | .014 | .005 | .013 | .013 | .001 | .001 |  |  |  |  |  |
| VB | .010 | .013 | .014 | .005 | .014 | .013 | .001 | .001 | 0 |  |  |  |  |
| JP | .013 | .015 | .016 | .009 | .015 | .015 | .006 | .006 | .006 | .006 |  |  |  |
| CH | .013 | .015 | .016 | .009 | .016 | .015 | .007 | .006 | .006 | .006 | .004 |  |  |
| FH | .076 | .074 | .074 | .072 | .074 | .074 | .072 | .072 | .072 | .071 | .074 | .069 |  |
| IC | .076 | .074 | .074 | .072 | .074 | .074 | .072 | .072 | .072 | .071 | .074 | .069 | 0 |

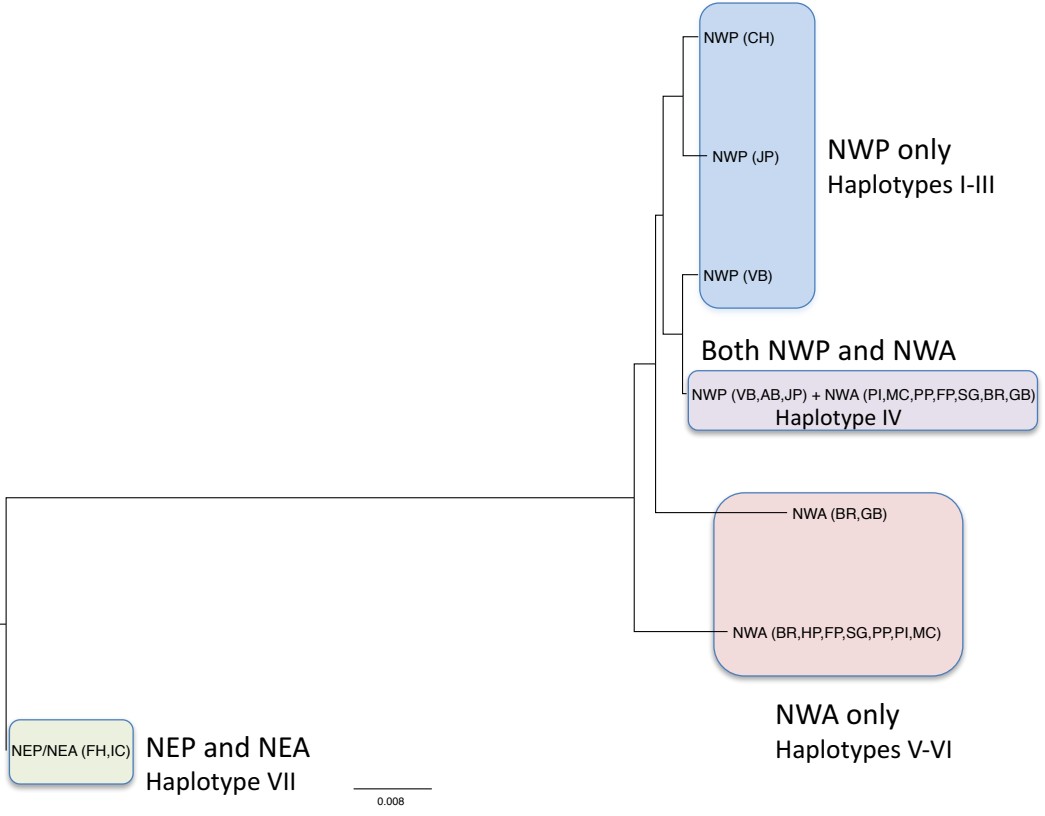

**Figure 2   Neighbor-joining tree of COI haplotypes based on Kimura 2-parameter distances.** The regions where the haplotypes are found are indicated.

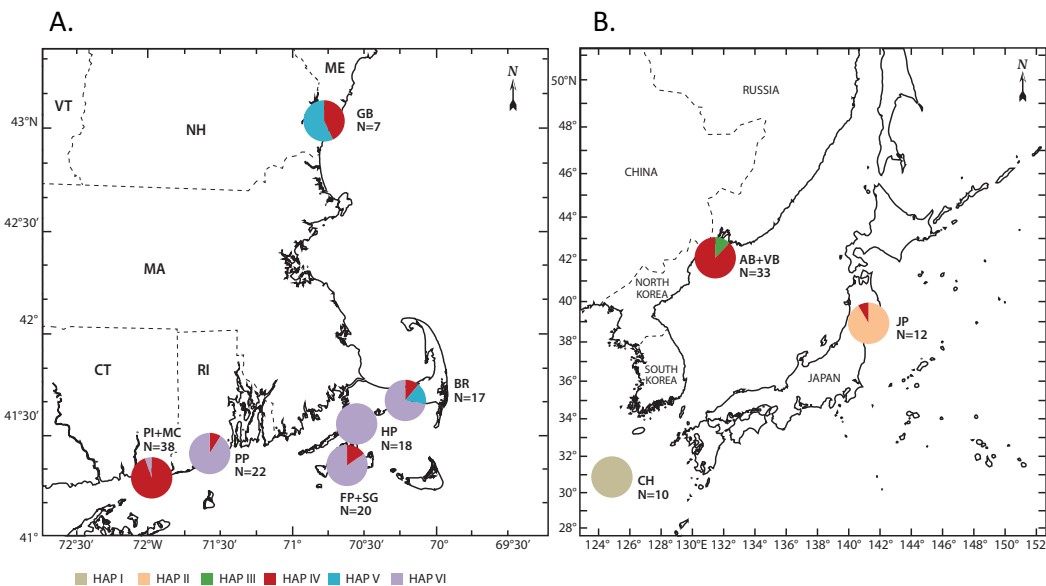

**Figure 3** **Geographic distribution of haplotypes.** (A) Distributions in the Northwest Atlantic; and (B) Distributions in the Northwest Pacific. Closely spaced sites are combined (Pine Island and Mumford Cove; Farm Pond and Sengekontacket Pond; Amur Bay and Vostok Bay). Individual site haplotype data are presented for these sites in Table 2 and Fig. 2. The CH haplotype presumably originates from the Chinese coast in the NWP but the location is not provided (unpublished Genbank entry) and so it is not depicted here. Site abbreviations are listed in Table 1 and haplotype numbers are as in Fig. 2.

Potter Pond locations, and absent in Hamblin Pond. Haplotype V was found in Great Bay and Bass River. Haplotype VI was found most commonly in Bass River, Farm Pond, and Potter Pond, and exclusively in Hamblin Pond. All NEP (San Juan Island) and the single NEA (Iceland) medusae shared an identical haplotype (Haplotype VII) that was not found in either the NWA or NWP.

The two unique NWA haplotypes (Haplotypes V and VI) were separated from each other by 12 nucleotide substitutions, and from the shared NWA/NWP (Haplotype IV) and unique NWP haplotypes (Haplotypes I–III) by 7–8 nucleotide substitutions. The NWP haplotypes (Haplotypes I–IV), including the one shared with the NWA, differed from each other by 1–3 nucleotide substitutions.

# DISCUSSION

## Population delineation and taxonomy

Our results refine, rather than resolve, the *Gonionemus* zoogeographic puzzle. Pairwise distances between the NEP/NEA *Gonionemus* and the NWP/NWA *Gonionemus* are typical for interspecific, rather than intraspecific, comparisons (*Ortman et al., 2010*; *Zheng et al., 2014*) and thus suggest the possibility that the two groups may represent different species. Cnidarian lineages differ substantially in their mitochondrial substitution rates (*Shearer et al., 2002*; *Govindarajan, Halanych & Cunningham, 2005*; *Wares, 2014*). And, based on a relative rate test on COI in the closely related *Maeotias marginata*, Limnomedusae may exhibit relatively slow rates (*Shearer et al., 2002*). Thus, the difference observed in

NEP/NEA and NWP/NWA forms could reflect especially deep divergences. It would be possible to calculate the substitution rate between the NEP/NEA and NWP/NWA forms if their divergence date could be reasonably estimated (e.g., from the opening of an Arctic passage, *Wares & Cunningham, 2001*; *Govindarajan, Halanych & Cunningham, 2005*); however, there is too much uncertainty at this point in the origin and dispersal history of *Gonionemus* as described below, to do this with confidence.

Species-level genetic divergence is corroborated by our incidental morphological observations, which are consistent with those from *Russell (1953)*. A future taxonomic re-assessment that incorporates morphology and multiple genetic markers could suggest that the NWP/NWA form should be reclassified as *G. murbachii* Mayer 1901. Because *G. murbachii* was first described in the NWA, this line of reasoning would suggest that *Gonionemus* could actually be native to the NWA. However, the situation is complex as our findings also refute previous taxonomic thought and do not necessarily dismiss an invasion event. We also cannot rule out the possibility that none of the contemporary NWA lineages correspond to *G. murbachii*. For clarity, we will refer to the contemporary NWA/NWP form as "*Gonionemus* sp." for the remainder of our discussion.

*Naumov (1960)* considered *G. vertens* and *G. murbachii* to be morphologically distinguishable subspecies. He described the Pacific form (including both NWP and NEP forms) as *G. vertens vertens*, and the Atlantic form as *G. vertens murbachii*. *Naumov (1960)* also noted that only the *G. vertens vertens* form is toxic to humans (although only in some Russian locations). Thus, while *Naumov (1960)* recognized two forms, as our results suggest, his delineation of those two forms differs substantially from ours. *Naumov (1960)* assumed each form was found in a different ocean, but our results show that both the *vertens* and *Gonionemus* sp. forms have disjunct distributions—with the *vertens* form in the NEP and NEA and the *Gonionemus* sp. form in the NWP and NWA. This is suggestive of anthropogenic dispersal in both forms. But again, the situation is complex, and possible explanations include both natural and anthropogenic causes. We now evaluate 4 possible invasion history scenarios in light of these results.

## NWA invasion scenarios
### Scenario 1 – "Traditional"
In the "traditional" scenario (e.g., *Tambs-Lyche, 1964*; *Edwards, 1976*; *Govindarajan & Carman, 2016*), the NEP/NEA and NWP/NWA forms are conspecific (=*G. vertens*), *G. vertens* would therefore be widely distributed in the Northeast and Northwest Pacific, where it may assume a range of morphological and toxicity phenotypes. Under this scenario, *G. vertens* was introduced multiple times from the Pacific to the NWA and NEA. NWA populations would stem from an introduction in the late 19th century of a less toxic variety (from the NEP directly or by way of the NEA where it is also assumed introduced), and a second introduction in the late 20th century of the highly toxic NWP variety. The observed variation in the different forms could be explained by environmental factors and not species differences—for example, *Miglietta & Lessios (2009)* showed that some hydrozoans can take on different phenotypes in different locations and thus confound the reconstruction of invasion history. However, our data strongly refute this scenario for

*Gonionemus*, as the high pairwise distance values between forms suggest that we are dealing with different species. Also, there are COI haplotypes in the NWA that are not shared with the NEP or NWP and are quite divergent themselves, and this could indicate a native NWA population (although this could also be due to relatively limited sampling in the NWP and NEP). Lastly, we found no NEP/NEA haplotypes in the NWA, despite extensive sampling near the locations of pre-1990 populations.

### Scenario 2—"Reverse invasion"

In this scenario, we assume the NWA/NWP *Gonionemus* is distinct from *G. vertens* in the NEP, and represents *G. murbachii*. The first record of the *G. murbachii* form in the NWP is from the 1920s, over 2 decades *after* it was described in the NWA (*Yakovlev & Vaskovsky, 1993*). Therefore, if we consider *G. murbachii* native to the NWA, its subsequent observations in the NWP indicate that it may have been introduced from the NWA to the NWP, *in the opposite direction* of what has been assumed. However, if the NWA form invaded the NWP, we might expect there to be reduced diversity in the NWP, and that diversity to be a subset of what is found in the NWA. But, although the resolving power of COI is limited, this is not what we observed. Rather, COI haplotype diversity is similar in the NWA and NWP, and there are unique haplotypes in both regions. Furthermore, NWP diversity is likely underestimated relative to the NWA, due to fewer sampling localities. And importantly, severe stings were recorded several decades in the NWP before they were recorded in the NWA—which, assuming the stings are heritable, strongly suggests an introduction in the direction of the NWP to the NWA (and not vice versa). Lastly, this scenario assumes that, if the NWA/NWP form is a different species than *G. vertens*, that it is *G. murbachii*, and not a third, undescribed, species. Analysis of historical specimens may help to resolve this last issue.

### Scenario 3—"Lineage admixture"

Here we also consider the NWA/NWP and NEP/NEA forms to be different species as in Scenario 2. The NWA and NWP forms are also different from each other, although they represent different lineages of a single species (*G. murbachii* or *Gonionemus* sp.) rather than different species. In this scenario we assume that the NWA contains at least 2 ostensibly native cryptic lineages (possibly segregated geographically, southern and northern New England; Fig. 3) that are distinct from each other and from the assumed native NWP lineages. There are also multiple NWP lineages (and likely more than what we observed with our limited sampling), only one of which is highly toxic (possibly represented by Haplotype IV, although mitochondrial haplotype should not be construed as equivalent to toxicity). The NWA experienced a single invasion (likely in the late 1980s) of the toxic NWP lineage, which is now interbreeding with the native NWA lineages. This hybridization may be facilitating NWA blooms and regional range expansion.

While a naturally disjunct distribution for *G. murbachii*/sp. seems unlikely, it could be explained by a trans-Arctic migration. Around $3\frac{1}{2}$ million years ago, a well-documented sea passage opened up connecting the north Pacific and the north Atlantic oceans (*Vermeij, 1991*). This passage enabled a large-scale migration of organisms, primarily in the direction of the North Pacific to the North Atlantic. Migrating seaweeds and seagrasses would likely

have carried associated fauna such as *Gonionemus*. More recent (e.g., in the Pleistocene; *Palumbi & Kessing, 1991*) or earlier (e.g., *Olsen et al., 2004*) trans-Arctic migrations are also potentially possible. The presence of unique haplotypes separated by several substitutions in both the NWA and NWP suggests independently diverging populations consistent with this explanation. As described above, the mitochondrial substitution rate might be particularly slow in Limnomedusae so that differences represent deeper divergences consistent with an earlier migration. It is also possible that multiple trans-Arctic migrations have occurred—for example, the first, resulting in the NEP/NEA and NWP/NWA split, and a second, resulting in the different NWP and NWA lineages.

An alternative variant of the ''Lineage Admixture'' scenario is that *G. murbachii*/sp. is native to the NWP, where it went unnoticed until the 20th century, and was introduced multiple times into the NWA, beginning in the late 19th century. This variant seems less likely, given the number of steps between the two unique NWA haplotypes to each other and to the NWP haplotypes. However additional sampling is necessary to definitively rule this out, and importantly, this variant still involves lineage admixture which has evolutionary and ecological implications as described below.

### Scenario 4—"Reverse admixture"

The distribution of Haplotype IV in the NWA appears to contradict the Lineage Admixture hypothesis in that it is most abundant in the northern and southern regions of the NWA range, and less abundant in the center, where the increase in NWA toxicity was first noticed. The first severe stings in the NWA that we are aware of were reported in 1990 in Waquoit Bay (which includes our Hamblin Pond site). But, we did not find the Vladivostok-area Haplotype IV at all in Hamblin Pond, and the frequency of Haplotype IV is greatest in the sites most distant from Hamblin Pond (Pine Island, Mumford Cove, Great Bay). It is possible that we did not find Haplotype IV in Hamblin Pond because it was missed in our sampling efforts or that Haplotype IV has declined in frequency due to selection (again remembering that mitochondrial haplotype is not equivalent to toxicity). It is also possible that *Gonionemus* stings in other parts of the NWA are unreported and so that the current outbreak did not begin in Waquoit Bay as the sting record in *Govindarajan & Carman (2016)* might suggest. However, we also consider a fourth scenario, where Haplotype IV represents the historical NWA *Gonionemus* populations. Here, the Haplotype IV form is probably an invader from the NWP. It is possible that despite an affinity with NWP toxic populations, painful stings did not occur because toxicity was tempered by (unknown) environmental factors. The recent outbreak of stings may indeed be caused by a new invasion, perhaps originating in the Waquoit Bay area, but the source population is unknown.

Our COI results provide new insight into the history of *Gonionemus* in the NWA but raise more questions than answers. Future studies should include sampling from several additional North Pacific and North Atlantic locations and utilize additional genetic markers like single nucleotide polymorphisms (SNPs) that can provide greater resolution than COI and indicate whether hybridization has occurred. Additionally, a better understanding of

the toxicity phenotype and its relationship with environmental triggers and genotype is crucial, both for understanding *Gonionemus* invasion history and for public health.

*Gonionemus* in all regions (NWP, NWP, NWA, NEA) may bloom episodically, and some populations may wax and wane, conceivably over the course of decades, due to environmental causes (*Condon et al., 2013*). Such periodicity could give the false impression of invasions. Both scenarios 2 and 3 challenge the long-standing assumption that *Gonionemus* was introduced in the NWA. When *Gonionemus* was first recorded in the NWA, one of the locations was in Woods Hole, MA, in a coastal pond adjacent to a marine laboratory (The Marine Biological Laboratory). The jellyfish quickly became the focus of several scientific studies. The long history of faunal studies in the Woods Hole region and the immediate attention that the jellyfish received after they were first recorded seems to support *Gonionemus*'s non-indigenous status in that region, because it was assumed local scientists would have seen it earlier if it were present. However, *Gonionemus* is capable of producing both asexual frustules and cysts, which can persist for unknown lengths of time (*Uchida, 1976*). In some species, cysts can potentially persist over decades (*Bouillon et al., 2004*). More study is required to understand how long these asexual stages can persist, the environmental triggers for their germination, and if they have played a role in NWA *Gonionemus* population dynamics.

### Evolutionary processes influencing NWA *Gonionemus*

Multiple evolutionary processes can influence colonizing populations, including introduced populations. It is often assumed that colonizing populations harbor only a subset of the genetic diversity found in parent populations. As genetic diversity is thought to promote population persistence, it follows that the low diversity in founding populations would make it difficult for them to become established. The fact that many colonizing populations do become established despite their assumed low diversity is called the "genetic paradox". Multiple inputs followed by lineage admixture may be a mechanism to overcome the "genetic paradox" (*Kolbe et al., 2004*; *Dlugosch & Parker, 2008*). In some cases, incoming populations may come from different source areas, and subsequent interbreeding could generate novel genetic combinations that also may help the nascent population become established (*Kolbe et al., 2004*).

Hybridization (or population admixture) may be an important mechanism leading to evolutionary change in NWA *Gonionemus* populations. This process could be occurring either between multiple anthropogenic inoculation events (e.g., Scenario 1) or between long-diverged intraspecific lineages (e.g., Scenarios 3 and 4). The resulting genetic changes could be promoting the prominent blooms and apparent rapid range expansion that has been observed since the 1990s. After the 1930s eelgrass dieoff, NWA *Gonionemus* was known primarily from a single pond on the island of Martha's Vineyard, Massachusetts, and very occasional reports from the Gulf of Maine (*Govindarajan & Carman, 2016*). These populations, which were apparently marginal for decades, may have been re-invigorated when mixed with recent NWP individuals, which provided the genetic material to enhance the jellyfish's fitness. Because our dataset is limited to a single mitochondrial marker, we cannot evaluate here if introgression or hybridization has occurred.

In contrast to the genetic paradox paradigm, the assumption that genetic diversity is low in colonizing populations may not hold true for many invading populations (e.g., *Darling et al., 2008*), and may be sufficient to allow for adaptive changes driven by natural selection (*Koskinen, Haugen & Primmer, 2002*; *Wares, Hughes & Grosberg, 2005*). Adaptive evolution could be responsible for differences in phenotype between invading and source populations in some species. Similarly, it is possible that contemporary NWA populations may be experiencing rapid adaptive evolution. However, detection of adaptation can be obscured by phenotypic plasticity (*Tepolt, 2015*; *Krueger-Hadfield et al., 2016*). Thus, future studies that use nuclear markers capable of finer-scale resolution and detection of hybridization, coupled with phenotypic characterization in Pacific and NWA regions, are necessary to assess both the lineage admixture and adaptive evolution hypotheses in *Gonionemus*.

## The toxicity phenotype

The toxicity phenotype stands out with special importance due to the concerns about envenomations, and it may also play a role in the NWA range expansion. *Gonionemus* medusae are predators that consume a variety of zooplankton prey (*Arai & Brinckmann-Voss, 1980*), and so increased toxicity could improve prey capture or deter predators and lead to population growth. The current NWA toxicity phenotype appears more aggressive than in pre-1990 populations, as evidenced by the history of stings to humans (*Govindarajan & Carman, 2016*). This observation could be due to lineage admixture between contemporary and historical NWA populations, recent adaptation in contemporary NWA populations, and/or plasticity due to environmental conditions.

The contemporary NWA toxicity phenotype could be subtly different than the NWP toxicity phenotype. In the NWP, especially off of Vladivostok, *Gonionemus* stings are a well-known hazard and there have periodically been mass-stinging events. In an extreme example, on just one day in June, 1966, over 1,000 people were stung by *Gonionemus* off of the Amur Bay recreational area (*Mikulich& Naumov, 1974*). Stings seem to be more common in the Vladivostok region in hot, dry years, suggesting environmental factors play a role. Also, cold winters associated with ice cover may be unfavorable for *Gonionemus* as a result of scour on the eelgrass beds and the pulse of fresh water when the ice melts (*Mikulich& Naumov, 1974*).

*Govindarajan & Carman (2016)* described a number of *Gonionemus* sting symptoms reported by victims including severe pain, respiratory difficulty, and temporary paralysis. It should be noted that these symptoms were based on the victims' personal accounts of the stings and not medical reports. Russian accounts describe additional symptoms including blindness and hallucinations (*Michaleff, 1974*; *Yatskov, 1974*) that have not been reported in the NWA as far as we are aware. However, because sting symptoms may be described differently by individuals, and individuals likely vary in their physiological sting responses, a toxicity assay that is objective and quantitative is necessary to accurately compare jellyfish toxicities. Given the apparent increase in *Gonionemus* bloom frequency and NWA range expansion, this is an area where more research is urgently needed.

The biogeography of *Gonionemus* has perplexed scientists for decades. New sightings in the NWA and recent reports of extremely painful stings speak to an urgent need for a better understanding of its origins, which can come from an approach that integrates its taxonomy, genetics, life cycle dynamics, and historical records (e.g., *Pringle et al., 2009*). While our study resolves some questions, it leads to many more new ones. We found that NEP *G. vertens* is distinct from the form found in the NWA and NWP. *Gonionemus* in the NWA and NWP may be *G. murbachii*, although a comprehensive taxonomic assessment is required to confirm the classification of this form. Our results show that both the NWA and NWP contain independent *Gonionemus* lineages, possibly as a result of trans-Arctic migrations. We also resurrect the hypothesis that *Gonionemus* could be native to the NWA, in contrast with the long held assumption that it was introduced by anthropogenic means. One lineage, which we term Haplotype IV, is shared with the well—known highly toxic populations in the coastal Vladivostok-area in the Sea of Japan. However, because the distribution of Haplotype IV does not correspond with the geographic distribution of NWA sting reports and mitochondrial lineage is not equivalent to toxicity, we cannot assume that this form was introduced recently. While it seems less likely, it is also possible that an invasion occurred in the reverse direction—from the NWA to the NWP. More research is needed to resolve these competing scenarios.

## ACKNOWLEDGEMENTS

We thank Bill Grossman and Dave Remsen (Marine Biological Laboratory), Dave Grunden (Shellfish Department, Town of Oak Bluffs, Massachusetts), Fred Short (University of New Hampshire), Charlie Woods (University of Connecticut), Claudia Mills (University of Washington), Alexei V. Chernyshev (A.V. Zhirmunsky Institute of Marine Biology, National Scientific Center of Marine Biology) and Hiroshi Miyake (Kitasato University) for assistance in collecting jellyfish and providing samples. We thank Óskar Sindri Gíslason (Southwest Iceland Nature Research Centre) for collecting the Icelandic specimen and Allen Collins (Smithsonian Institution) for obtaining its COI sequence. We thank Lubov Petrova (Primorsky Aquarium) and Eric Lazo-Wasem (Yale Peabody Museum) for supplying *Gonionemus* photographs. We thank Ferdinando Boero and Cinzia Gravili (Università del Salento) for sharing their literature collection. We thank Pam Polloni (Woods Hole Oceanographic Institution) for laboratory and proofreading assistance.

### Funding

This work was supported by the Woods Hole Sea Grant, the Town of Oak Bluffs Community Preservation Committee, the Nantucket Biodiversity Initiative, the Kathleen M. and Peter E. Naktenis Family Foundation, and the Russian Science Foundation (No. 14-50-00034). The funders had no role in study design, data collection and analysis, decision to publish, or preparation of the manuscript.

## Grant Disclosures

The following grant information was disclosed by the authors:

Woods Hole Sea Grant.

Town of Oak Bluffs Community Preservation Committee.

Nantucket Biodiversity Initiative.

Kathleen M. and Peter E. Naktenis Family Foundation.

Russian Science Foundation: 14-50-00034.

## Competing Interests

The authors declare there are no competing interests.

## Author Contributions

- Annette F. Govindarajan conceived and designed the experiments, performed the experiments, analyzed the data, contributed reagents/materials/analysis tools, wrote the paper, prepared figures and/or tables, reviewed drafts of the paper.
- Mary R. Carman conceived and designed the experiments, performed the experiments, contributed reagents/materials/analysis tools, reviewed drafts of the paper.
- Marat R. Khaidarov performed the experiments, contributed reagents/materials/analysis tools, reviewed drafts of the paper.
- Alexander Semenchenko performed the experiments, reviewed drafts of the paper.
- John P. Wares analyzed the data, reviewed drafts of the paper.

## Data Availability

The mitochondrial COI sequences presented here are deposited in GenBank (accession numbers KY437814–KY437985).

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
