# Peer review of "Mitochondrial diversity in Gonionemus (Trachylina:Hydrozoa) and its implications for understanding the origins of clinging jellyfish in the Northwest Atlantic Ocean"

_PeerJ, doi:10.7717/peerj.3205_

## Round 0.1 · original submission · Minor Revisions

I have received comments from two reviewers. Both are very positive about your subject matter and work, with one recommending outright acceptance, and the other with some constructive comments that can help improve your paper. In particular, I agree with reviewer 2 that discussion about "origin" and "invasion history" are not clearly definable or obtainable by your results. Instead, your contribution seems to be more on “genetics related to population identity and structure”, and I hope you can restructure your paper (and title) in such a manner to reflect the slightly more speculative nature of the work.

Based on this, my decision is "major revisions" are needed. I look forward to receiving a new version.

·

Basic reporting

Clear, unambiguous, professional English, language used throughout: yes

Intro & background to show context: yes

Literature well referenced & relevant: yes, excellent, including difficult to access papers

Structure conforms to PeerJ standards: yes

Figures are relevant, high quality, well labelled & described: yes

Raw data supplied (see PeerJ policy): yes (Genbank)

Experimental design

Original primary research within Scope of the journal: yes

Research question well defined, relevant & meaningful. It is stated how the research fills an identified knowledge gap: yes

Rigorous investigation performed to a high technical & ethical standard: yes

Methods described with sufficient detail & information to replicate: yes

Validity of the findings

Impact and novelty not assessed: is assessed

Negative/inconclusive results accepted: inconclusive results are clearly indicated

Meaningful replication encouraged where rationale & benefit to literature is clearly stated: limited by available/accessible material

Data is robust, statistically sound, & controlled: cannot be evaluated, but best possible sampling with available resources was done

Conclusions are well stated, linked to original research question & limited to supporting results: yes

Speculation is welcome, but should be identified as such: no speculations, but differing hypothesis are presented

Additional comments

Phylogeographic studies on hydromedusae are desperately missing and the present study is thus a highly welcome trail-blazer publication with a high impact potential. Moreover, it concerns a serious stinger jellyfish and a species with a long-standing taxonomic discussion. Although more data are clearly needed for more robust conclusions, the study will certainly attract a wide audience and incite other researchers to contribute additional data from other populations.

·

Basic reporting

The article has minors considerations regarding main objective, results and final discussion/conclusion.

Overall it is a very well written article, with adequate references, background and general structure.

Experimental design

The article has a proper design dealing with genetics and marine populations considering a very peculiar organism.

Validity of the findings

Findings are very interesting and well presented; speculative scenarios are well identified in most of the article (minor corrections are suggested).

---

## Round 0.2 · accepted · Accept

The manuscript has been well refined, and both the reviewer and myself feel this version is acceptable for publication. If you wish, please consider a simplification of the title as suggested by the reviewer. I look forward to seeing the published version of this work!

·

Basic reporting

All considerations accomplished.

Experimental design

All considerations accomplished.

Validity of the findings

All considerations accomplished.

Additional comments

The new version has important improvements considering format (figures and tables), text and general conclusions. As we (reviewers) commented in the first round of review, this will be an important double contribution for marine biology: for biogeographical studies (considering regions in general), and for hydrozoan genetics&population studies in particular (indeed, very needed).

I just recommend to simplify the title to (avoid to repeat "clinging jellyfish"): "Mitochondrial diversity in <Gonionemus> (Trachylina:Hydrozoa) and its implications for understanding the origins of clinging jellyfish in the Northwest Atlantic Ocean"